# Template-Independent Poly(A)-Tail Decay and RNASEL as Potential Cellular Biomarkers for Prostate Cancer Development

**DOI:** 10.3390/cancers14092239

**Published:** 2022-04-29

**Authors:** Gordana Kocić, Jovan Hadzi-Djokić, Andrej Veljković, Stefanos Roumeliotis, Ljubinka Janković-Veličković, Andrija Šmelcerović

**Affiliations:** 1Department of Biochemistry, Faculty of Medicine, University of Niš, 18000 Niš, Serbia; andrej.veljkovic@medfak.ni.ac.rs; 2Serbian Academy of Sciences and Arts, 11000 Belgrade, Serbia; jovanhdj@sanu.ac.rs; 3Division of Nephrology and Hypertension, 1st Department of Internal Medicine, AHEPA Hospital, School of Medicine, Aristotle University of Thessaloniki, 541 24 Thessaloniki, Greece; roumeliotis@auth.gr; 4Department of Pathology, University Clinical Center Niš, 18000 Niš, Serbia; ljubinka.jankovic.velickovic@medfak.ni.ac.rs; 5Department of Chemistry, Faculty of Medicine, University of Niš, 18000 Niš, Serbia; andrija.smelcerovic@medfak.ni.ac.rs

**Keywords:** prostate cancer, poly(A) deadenylase, RNASEL

## Abstract

**Simple Summary:**

The ultimate need in cancer tissue is to adapt translation machinery to accelerated protein synthesis in a rapidly proliferating environment. Our study was designed with the aim of integrating fundamental and clinical research to find new biomarkers for prostate cancer (PC) with clinical usefulness for the stratification prediction of healthy tissue transition into malignant phenotype. This study revealed: (i) an entirely novel mechanism of the regulatory influence of Poly(A) deadenylase in mRNAs translational activity and the 3′ mRNA untranslated region (3′UTR) length in cancer tissue and its regulation by the poly(A) decay; (ii) the RNASEL interrelationship with the inflammatory pattern of PC and corresponding tumor-adjacent and healthy tissue; and (iii) the sensitivity, specificity, and predictive value of these enzymes. The proposed manuscript is based on the use of specific biochemical and immunoassay methods with the principal research adapted for the use of tissue specimens.

**Abstract:**

The post-transcriptional messenger RNA (mRNA) decay and turnover rate of the template-independent poly(A) tail, localized at the 3′-untranslated region (3′UTR) of mRNA, have been documented among subtle mechanisms of uncontrolled cancer tissue growth. The activity of Poly(A) deadenylase and the expression pattern of RNASEL have been examined. A total of 138 prostate tissue specimens from 46 PC patients (cancer specimens, corresponding adjacent surgically healthy tissues, and in their normal counterparts, at least 2 cm from carcinoma) were used. For the stratification prediction of healthy tissue transition into malignant phenotype, the enzyme activity of tumor-adjacent tissue was considered in relation to the presence of microfocal carcinoma. More than a four-times increase in specific enzyme activity (U/L g.prot) was registered in PC on account of both the dissociation of its inhibitor and genome reprogramming. The obtained ROC curve and Youden index showed that Poly(A) deadenylase identified PC with a sensitivity of 93.5% and a specificity of 94.6%. The RNASEL expression profile was raised significantly in PC, but the sensitivity was 40.5% and specificity was 86.9%. A significantly negative correlation between PC and control tissue counterparts with a higher expression pattern in lymphocyte-infiltrated samples were reported. In conclusion, significantly upregulated Poly(A) deadenylase activity may be a checkpoint for the transition of precancerous lesion to malignancy, while RNASEL may predict chronic inflammation.

## 1. Introduction

Prostate cancer (PC) represents a leading cause of cancer-related deaths in men. Known contributing risk factors include old age (50+), race (African American), family history, diet (meat and dairy products), and chronic prostate infections. Different pathogen-associated molecular patterns (PAMP) and damage-associated molecular patterns (DAMP) may induce inflammation, such as bacterial infections, viruses, nutrients, hormones, urine reflux, or autoimmune reactions [1,2]. Along with the study of the subtle molecular mechanisms of carcinoma development, there have been efforts to develop early and reliable diagnostic markers [2,3].

Besides the excessive DNA replication, cancer growth is a consequence of transcriptional deregulation. It has been almost 30 years since Cox and Goding posed a promising hypothesis: “transcription-related research should soon yield major dividends for cancer patients” [4]. Afterwards, the mechanisms of RNA transcription, maturation, and turnover were documented and became mainstream, starting a new trend in cancer research.

The cell-cycle progression, mitotic division, meiotic maturation, embryogenesis, differentiation, and cell response to exposomic factors occur by the modulation of mRNA stability and protein translation machinery. The post-transcriptional messenger RNA (mRNA) maturation, decay, and turnover rate are regulated by the half-life of the template-independent poly(A) tail, localized at the 3′-end of the 3′-untranslated region (3′UTR) of mRNA. In normal cells, remaining in a non-proliferative state, it represents a highly conserved mechanism, a highway of mRNA degradation; afterwards, the mRNA can undergo decapping [5,6]. It is catalyzed by poly(A) deadenylases [7,8]. Minor pathways of decay may be the deadenylation-independent decapping and endoribonucleotidase-catalyzed degradation of mRNA. Besides this, mRNAs can be degraded by a nonsense-mediated decay, a type of accelerated degradation aimed to reduce potential errors in gene expression [9,10]. 

The most recent approaches that have changed previous paradigms about the decay of 3′UTR poly(A) tail include the following: shorter poly(A) nucleotide tracks and longer mRNA stability, specifically observed in malignant tissues; protein synthesis translated from the shorter mRNA is several times higher than that from the longer mRNA; shorter poly(A) tail mRNA isoforms are capable of producing several times more proteins than the longer ones; mutations resulting in longer poly(A) tracks reduced the protein synthesis rate and mRNA stability; 3′UTR shortening in the mRNAs of proto-oncogenes, which led to their transformation into oncogenic proteins, translated without repressive control of miRNAs [11,12,13]. In this way, the diminishing of the “polyadenylation code”, known as the “survival of the fittest”, represents a functional adaptation of malignant cells to escape translational control, by the malignant cell demand. A potential mechanism of their oncogene action has been explained through the loss of microRNA (miRNA) complementary binding sites on mRNAs, which are usually located in the 3′UTR region [14]. 

Concerning the compartmentation of PC inside of the gland, the peripheral zone (PZ) represents the typical localization for cancer and inflammation. That is why it is not surprising that chronic infection and inflammation very often coexist with cancer. Inflammation may induce tumor growth by causing DNA damage, usually associated with inadequate DNA repair. On the other hand, inflammation may induce the activation of immune defense cells, which may protect tissue from unwanted cells bearing damaged DNA [15,16]. Among diagnostic values, a systemic immune-inflammation index (SII) and neutrophil–lymphocyte ratio (NLR) have been documented [17]. Recent findings suggest that chemokines and cytokine-mediated signaling pathways are intensively involved in PC growth, angiogenesis, endothelial mesenchymal transition, leukocyte infiltration, and hormone resistance in advanced types of PC [18]. Recently, Ribonuclease-L (RNASEL, 2′,5′-oligoisoadenylate synthetase-dependent) has been the subject of intensive research, as the key RNase in a viral RNA decay and inflammation. It triggers the synthesis and secretion of inflammatory cytokines, particularly type I interferon (IFN) [19,20]. The role of RNASEL in hereditary prostate cancer 1 (HPC1) has an intriguing significance. Hereditary mutations in RNASEL may predispose an increased incidence of PC and may determine the aggressiveness of the disease [21,22,23,24,25]. Catalytic action of RNASEL may also produce small non-coding double-stranded RNAs (dsRNAs), important regulators of cell survival via autophagy, versus apoptosis [26]. The role of pleiotropy in PC makes it attractive and currently still mysterious.

The primary focus of our research is the poly(A) decay as a potential checkpoint in PC development and progression. In order to employ the Poly (A) deadenylase as a possible marker of healthy tissue transition into malignant phenotype, it has been detected in carcinoma tissue, adjacent surgically healthy tissue, and in their normal counterparts, at least 2cm from carcinoma. In the present study, we also compared the expression level of RNASEL, in the same tissue specimens. Standard markers of the PC stage and progression (PSA, Gleason Score, and histopathological specimens) were also evaluated. Considering these enzymes as proteins with targeted non-coding RNA regions or non-coding RNAs as substrates, they may represent novel RNA–protein by-pass cellular biomarkers.

## 2. Patients and Methods

The Research Ethics Committee at the Faculty of Medicine Nis approved this prospective study protocol and waved informed consent (N^o^12-8818-2/18 on 23 September 2020).

Patient selection: Our pilot study was conducted at University Clinical Center Nis, with 46 consecutive patients with prostate cancer (PC) who underwent radical prostatectomy. The diagnosis was verified by clinical symptoms, abnormal findings on digital rectal examination (DRE), and an increased age-specific reference range of PSA value.

Tissue preparation: After prostatectomy, the parts of the cancer tissue, adjacent surgically healthy tissue, and normal tissue counterpart, at least 2cm from carcinoma, were dissected. The samples obtained were homogenized on ice; 10% of the homogenates were prepared and frozen on −80 °C until the biochemical examinations were performed.

Enzyme assays: The protocol used for the determination of the activity of Poly(A) deadenylase was optimized in our laboratory, previously published for tissue and cell culture samples and for plasma specimens [27,28,29]. The method was based on spectrophotometric measurement of released acid-soluble nucleotides at 260 nm, from homopolynucleotide poly(A) as the substrate (purchased from Sigma-Aldrich, Darmstadt, Germany). Enzyme activity was expressed as the total (U/L homogenate) and specific enzyme activity (U/g protein of fresh tumor tissue). For the stratification method for the evaluation of poly(A) deadenylase as a predictive marker of healthy tissue transition into malignant phenotype, the enzyme activity of tumor-adjacent tissue was considered in relation to whether there was a microfocal carcinoma or not.

To evaluate the possible predictive ability of Poly(A) deadenylase-specific activity for identifying PC, we performed receiver operation curves (ROC) and then calculated the Youden index, to determine the optimal cutoff value of Poly (A) deadenylase-specific activity. We then calculated the sensitivity and specificity to predict prostate cancer.

Having in mind the importance of RNase inhibitors for limited cell RNase activity, the activity of latent, i.e., inhibitor-bound RNase, was estimated [30]. The dissociation of Poly(A)deadenylase/inhibitor complex was achieved by using sulfhydryl reagent (0.1mL of 10mM p-chloromercuribenzoate) prior to the determination of enzyme activity [31]. In this way, the enzyme activity was calculated as (i) total (free + inhibitor bound); (ii) free; and (iii) latent, i.e., inhibitor-bound.

The protocol for RNASEL (2′,5′-Oligoisoadenylate Synthetase-Dependent) was based on enzyme-linked immunosorbent assay (kits were purchased from Cloud-Clone Corp., Katy, TX, USA) with a detection range between15.625 and 1000 pg/mL. The specific enzyme expression was calculated according to the tissue protein content (ng/g proteins).

To evaluate the possible predictive ability of the RNASEL-specific expression pattern for identifying PC, we performed receiver operation curves (ROC) and then calculated the Youden index to determine the optimal cutoff value of RNASEL-specific expression. Then, we calculated the sensitivity and specificity to predict the presence of PC.

In order to distinguish enzyme activity in relation to inflammatory conditions, the enzyme activity was considered in relation to whether there was predominantly lymphocyte, macrophage–neutrophil inflammation, or only tissue hypertrophy.

The tissue protein content in homogenates was measured according to the Lowry procedure [32].

Statistical analyses: the results obtained are expressed as the mean ± standard deviation for continuous variables. Data analysis was performed using SPSS (one-way ANOVA) test. To determine the strength of a possible interaction and to quantify a possible association between two variables, a bivariate Pearson correlation coefficient was determined.

## 3. Results

Clinical characteristics of patients: The clinical characteristics of patients and the level of standard biomarkers are shown in Table 1.A prostate-specific antigen (PSA) test was used to diagnose PC, in which values above 4 ng/mL were suspicious for cancer.

Enzyme assays: Poly(A) deadenylase and RNASEL were measured in PC tissue, adjacent surgically healthy tissue, and in corresponding healthy counterparts, at least 2 cm from carcinoma. The corresponding samples were considered further in relation to a possible influence of tissue transition into malignant phenotype and inflammation; the samples were further subdivided in the corresponding groups.

Poly(A) deadenylase: More than a four-times increase in specific enzyme activity (U/L g.prot) was registered for Poly(A) deadenylase, followed by a more-than twofold increase in its activity in adjacent carcinoma tissue, compared to the control healthy tissue counterparts (Figure 1).

Our preliminary results showed no overlapping value between the PC and control samples, which maybe a prerequisite for considering sensitivity and specificity as well as the cutoff value in a larger series of samples. The enzyme activities for the corresponding prostate tissue specimens for each patient were evaluated with regard to their biomarker potential to distinguish cancer tissue, as a prognostic biomarker of cancer aggressiveness, and as potential predictive biomarkers for the stratification of transition of benign hyperplasia to the malignant process. Following the stratification of benign hyperplasia (BPH) and the appearance of microfocal cancer, apart from the main carcinoma tissue, the appearance of microfocal cancer was registered in 45.65 tumor-adjacent specimens. The enzyme activity was significantly higher in the adjacent tissue with microfocal carcinoma compared to tumor-adjacent tissue bearing only BPH and control specimens but was still significantly lower than that of PC specimens. Poly(A) deadenylase may be considered as an early marker for the transition of benign hyperplasia to a malignant one, when histopathological diagnosis is still insufficient. The evaluation of areas under the curves (AUCs) showed that Poly(A) deadenylase-specific activity (AUC = 0.97, 95% CI = 0.95–1.00, *p* < 0.0001) (Figure 2) predicted PC, and specific activity exhibited a significantly high performance. After determining the optimal cutoff values by Youden’s index, we calculated the sensitivity and specificity and found that Poly(A) deadenylase identified prostate cancer with a sensitivity of 93.5% and a specificity of 94.6%.

In monitoring the potential specificity of enzyme activity with regard to inflammatory conditions, the histopathological findings of prostatitis caused by infiltration of the prostate tissue by immune cells (lymphocytes, macrophages, or neutrophils) did not have any influence on enzyme activity. It may exclude any inflammatory process as a confounding condition for the increased Poly(A) deadenylase activity.

The dissociation of Poly(A) deadenylase-inhibitor complex by *p*-chloromercuribenzoate indicated that the main part (59.16%) of Poly(A) deadenylase in control healthy tissue seems to be latent: the inhibitor-bound. No quantity of latent form was detected in PC. The free enzyme in PC specimens was still more than 50% (51.6) higher than the total activity in the control tissue, which may indicate that about 50% of enzyme activity was raised because of genome reprogramming and the consequent increased expression in malignant tissue. Unlike malignant tissue (PC specimens), in tumor-adjacent tissue, only a gradual dissociation of enzymes from its inhibitor complex was documented since the latent form was retained in 35.23% in tumor-adjacent tissue and only in 5.11% in tumor-adjacent tissue with microfocal cancer. Based on the results obtained, it can be assumed that the increased expression of the enzyme may be a checkpoint for the transition to a malignant phenotype (Figure 3).

RNASEL: A significant difference in RNASEL in investigated groups of samples, concerning the total and specific expression pattern, was observed (Figure 4). The statistical significance was reported in PC specimens only for the total enzyme activity. Since there was no difference in RNASEL concerning the presence of microfocal lesions, tumor-adjacent tissue was not stratified. Based on these results and the difference obtained between the RNASEL expression profile in PC in relation to the control tissue, it would not be considered as an early tumor marker.

Evaluation of areas under the curves (AUCs) showed that RNASEL specific expression (AUC = 0.64, 95% CI = 0.53–0.74, *p* = 0.013), (Figure 5) predicted PC. However, the predictive ability of RNASEL was only modest. After determining the optimal cutoff values by Youden’s index, we calculated the sensitivity and specificity, which were 40.5% and 86.9%, respectively.

The on–off switch negative correlation was reported between RNASEL in carcinoma specimens and healthy tissue, since high activity in healthy tissue was followed by a fall in carcinoma tissue, and vice versa (Figure 6), where the correlation coefficient was −0.5.

In monitoring the type of inflammation as the confounding condition for the RNASEL expression profile, the RNASEL was stratified according to the type of inflammation (predominantly chronic lymphocyte infiltration, macrophage/neutrophil infiltration, or the absence of inflammatory cells). Although lymphocytic infiltration tended to be associated with higher RNASEL, it was statistically significant only in control specimens, compared to macrophage/neutrophil infiltration or the absence of inflammatory cells (Figure 7).

The pie charts in Figure 5 explore the percentage influence of lymphocyte infiltration, macrophage/neutrophil infiltration, or the absence of inflammatory cells in different tissue specimens. By analyzing the tissue slices, it seems important to note that only 37% of control samples showed marked inflammation, which decreased in tumor-adjacent tissue to 30% but moderately increased in PC specimens to 54%.

Examples of tissue histopathological findings in the above-mentioned specimens are documented in Figure 8.

## 4. Discussion

In our study, the Poly(A) deadenylase and RNASEL were determined in PC tissue, adjacent surgically healthy tissue, and in corresponding healthy counterparts, at least 2 cm from carcinoma. The increase in poly(A) deadenylase-specific activity ranged from two to ten times, followed by a more-than twofold increase in its activity in adjacent carcinoma tissue, compared to the control healthy tissue counterparts (Figure 1). The preliminary results obtained may consider Poly(A) deadenylase as a potential surrogate marker for transition of hypertrophic tissue into malignant one, so it is worth paying attention to sensitivity and specificity as well as to cutoff value in a larger series of samples.

From the first understanding of PC development and progression, there has been a tendency to define and establish an ideal or at least an early tumor marker, which would have a key or profound impact on pathogenesis of prostate cancer, early diagnosis, and possible management [1,2]. To define a reliable prostate tumor marker, the National Institute of Standards and Technology (NIST), the American Cancer Society National Prostate Cancer Detection Project, and other relevant associations tried to define criteria for clinical and laboratory prostate tumor marker assessment. According to the biological structure, cancer biomarkers are currently classified as DNA-based, RNA-based, and protein-based. Apart from biochemical structure and cell function, to consider any biomarker for practical clinical use, it should have high diagnostic specificity for detection and for monitoring the stage and prognosis; to be reliable in clinical intervention, recurrence, and survival; to be easily measurable in biological fluids (plasma, urine);and to be inexpensive. So far, more than fifty biologically active molecules have been identified with more or less proper significance in cancer development, staging, and reaction to therapy and overall survival. To promote one biomolecule as a potential biomarker, the first step is laboratory validation of the method and correlation of clinical significance with standard biomarkers. As routine diagnostic tests for PC development and progression, a prostate-specific antigen (PSA) test and digital rectal examination (DRE) [33,34] are proposed. In the current literature, we did not find any report about Poly(A) deadenylase activity in PC cells at the time this article was prepared. The results obtained about a manifold increase in cancer tissue and in tumor-adjacent tissue may propose Poly(A) deadenylase as a new and potential tumor marker for PC and a possible diagnostic marker for the transition from normal prostate tissue to cancer growth (Figure 1). To evaluate the possible predictive ability of Poly(A) deadenylase-specific activity for identifying PC, the receiver operation curves (ROC) and the Youden index determined the optimal cutoff value. We found that Poly(A) deadenylase identified PC with a sensitivity of 93.5% and a specificity of 94.6% with no overlapping values (Figure 2). Gene reprogramming in cancer occurs through differential expression of cancer-related genes, which leads to differential quantity of specific proteins in cancer tissue. The difference obtained between inhibitory bound and free enzyme activity in PC specimens and control samples may highlight a significant protein reprogramming pattern in prostate carcinoma owing to Poly(A) deadenylase, responsible for at least 50% of its expression (Figure 3).

The active mRNA transcript synthesis occurs through the final structuring of the proper poly(A) tail length. The proper poly(A) tail allows nuclear processing of mRNA and translation initiation after binding to poly(A)-binding protein (PABP). From the other side, the shorter 3′UTR poly(A) tail may diminish the posttranscriptional gene regulation, because they are targets for translational inhibition and mRNA destabilization, assembling into RNA-induced silencing complex (RISC). The rapid deadenylation process may represent a defense system against an aberrant or “unfavorable” mRNAs persistence, which have been found in cancerous tissue [12,13]. Experimental knockout of poly(A) deadenylase gene in a culture of gastric cancer cells resulted in the cell-cycle arrest of G0/G1 phase, followed by the accumulation of p21 tumor suppressor protein [35]. A high expression pattern of Poly(A) deadenylase has been documented in acute leukemias [36]. With regard to specific families, two main families of poly(A) deadenylases were isolated: the DEED types and the exonuclease–endonuclease–phosphatases (EEP) types. The Poly(A)-specific ribonuclease (PARN), POP2 endonuclease, CAF1Z and PAN2 members belong to the DEED family, while CCR4, Nocturinin, ANGEL and 2′ phosphodiesterase (2′PDE) belong to the EEP family [37,38,39,40]. Recently, only PARN has been proposed as a potential target of experimental cancer treatment [41]. Trascriptomic analysis of Poly(A) deadenylase expression in squamous cell lung carcinoma referred only PARN and Nocturnin(NOC) type of Poly(A) deadenylase family, significantly over-expressed, with a significant prognostic value in specific subtypes [42]. In this way, Poly(A) deadenylase may represent an integrative part of the cell cycle and survival control checkpoint in tumorigenesis. Aberrant synthesis of many functional “checkpoint” proteins necessary for cell proliferation, together with the synthesis of mutant forms of tumor suppressor proteins, has been documented in cancer pathogenesis [43]. In highlighting the specific role of Poly (A)-specific ribonucleases in reproductive tissues, a specific function in male spermatogenesis, female oogenesis, and fertility was documented [44,45]. Mice with the loss of PARN-regulatory protein (Cnot7-knockout mice), besides compromised deadenylation, suffer from sterility because of oligo-astheno-teratozoospermia and defective maturation of spermatids [46]. These data may point to the crucial importance of the poly(A) tail deadenylation regulation in the reproductive system and prostatic gland function.

In our study, the dissociation of Poly(A)deadenylase-inhibitor complex indicated that the main part (59.16%) of Poly(A) deadenylase in control healthy tissue seemed to be latent: the inhibitor-bound. Since the total activity of enzyme in PC specimens was 51.6% higher than the total activity in the control tissue, it may be calculated that about 50% of enzyme activity was raised because of genome reprogramming and increased expression in malignant tissue. At the same time, it is important to note that the tumor-adjacent tissue may behave like tissue at the crossroads between healthy and malignant phenotype. The appearance of microfocal carcinoma is followed by gradual liberation of the enzyme from its inhibitor complex (from 35.2 to 5.11% of inhibitor-bound enzyme) but still appears to have no effect on tissue reprogramming (Figure 3). The mechanism of enzyme–inhibitor binding has been documented in detail. RNases may make the noncovalent complexes with its natural protein inhibitor in the cell. The proposed roles of RNase inhibitor are the protection, control, and termination of cellular RNA degradation, hence the name “RNAguard”. RNase inhibitor proteins are ubiquitous, meaning that they usually follow RNase localization. Regarding the primary structure of inhibitor, it contains highly repetitive leucine-rich amino acid sequences and about 30 reduced cysteine residues of the 32 available. Regarding the tertiary structure, it is made of α-helix and β-strand. Its tertiary horseshoe-like structure, rich in leucine residues, may allow for the tight equimolar interaction with the enzyme [47,48]. The importance of inhibitors in the germinal organs of men is evidenced in the fact that, besides the brain and liver, the testicular germ cells are very rich in RNase inhibitor [49,50,51,52]. Due to a large quantity of reduced cysteine, the RNase inhibitor can be inactivated by SH group-modifying reagents, such as PCMB (p-chloromercuriobenzoate), which induces the dissociation of free enzyme and inactive inhibitor [50]. In our earlier results, we documented the influence of steroid hormones on RNase-inhibitor dissociation [53]. Decreased binding of RNase inhibitor was documented in leukemia [54]. In order to explain the potential mechanism of latent enzyme release from the inhibitory complex in PC, we have considered our recent results in relation to the increased generation of free radicals in prostate carcinogenesis, presumably owing to the increased xanthine oxidase/dehydrogenase ratio [55]. Liberated free radicals may oxidize cysteine SH groups in the protein inhibitor, which play a structural and functional role in inhibitor function. Their oxidation can induce conformational changes of inhibitor and can induce the dissociation of enzyme-inhibitor complex [47]; the oxidation of SH groups is a mechanism of in vitro dissociation of latent (inhibitor-bound) enzyme in order to measure latent enzyme.

The mean values of RNASEL in tumor samples were significantly raised, as shown in Figure 4. Evaluation of areas under the curves (AUCs) showed that RNASEL-specific expression may predict PC, but with a low sensitivity of 40.5% and a specificity of 86.9% (Figure 5). A highly negative correlation in cancer vs. corresponding healthy tissue is documented (Figure 6). The possible reason why RNASEL expression was highly negatively correlated in carcinoma tissue to its normal counterpart may be found in the immune-suppressive properties of RNASEL. Besides viral RNAs, RNASEL may initiate the cleavage of other cellular RNAs, which may promote cell apoptosis [20,24,26,56]. Because of apoptotic properties, RNASEL has been proposed as a tumor-suppressor molecule [57,58]. Immunosuppressive properties as the result of increased RNASEL in cancer tissue may be explained as a possible immune-escaping mechanism of cancer tissue, which stands in opposition to more pronounced inflammation. Inflammation as an epigenetic factor may have an influence on DNA damage or the aberrant expression of cell-cycle control proteins [15,16,17,18]. It was documented that the type of infiltrates made of inflammatory, innate immune cells and CD4+ T-lymphocyte, may predict cancer progression. The cells of innate immunity, such as macrophages and immune suppressor cells, may predict prostate cancer progression, while surrounding with adaptive immunity cells may act as tumor suppressive cells [56,57,58]. RNASEL represents a specific type of “housekeeper enzyme”, the key switching anti-infective mechanism activated immediately after viral attack, or interferon-receptor binding, through the short 5′-phosphorylated, 2′,5′-linked oligoadenylate, known as the 2-5 oligoadenylate system (2-5A). Once activated by 2-5A, the inactive monomeric RNASEL makes a complex 2-5A/RNASEL system, which is responsible for the cleavage of single-stranded regions of RNA, located near the UpUp or UpAp dinucleotides, or double-stranded RNAs, which are typical viral PAMP molecules (pathogen-associated molecular pattern) [59]. The hypothesis has been corroborated by the specific location of the *RNASEL* gene at the chromosome region 1q25, specifically susceptible to rearrangement in some cancer types. Missense point mutations of *RNASEL* gene resulting in aberrant RNASEL structure (arginine to glutamine substitution at position 462) followed by defective function were documented in some families with hereditary prostate cancer [23,24,25,26]. The specific Q variant of RNASEL with an almost threefold lower catalytic activity has been registered in about 13% of patients with carcinoma of the prostate, which may be accounted for by the increased risk of prostate cancer in about 50% in the case of heterozygous mutation, while its appearance in a homozygous form may increase the prostate cancer risk two times. Besides a genetic variant, the epigenetic alteration of RNASEL catalytic activity influenced by different inflammatory or infective agents would not be excluded in prostate cancer [19,21,22,23,24,25]. In our study, the statistical significance was reported in PC specimens only for total RNASEL. In monitoring the type of inflammation as the confounding condition for RNASEL expression profile, the RNASEL was stratified according to the type of inflammation (predominantly chronic lymphocyte infiltration, macrophage/neutrophil infiltration or the absence of inflammatory cells). Although lymphocytic infiltration tended to be associated with higher RNASEL, it was statistically significant only in control specimens, compared to macrophage/neutrophil infiltration or the absence of inflammatory cells (Figure 7). The percentage contribution and profile of immune cells infiltration is also shown in Figure 7 in the pie charts. Only 37% of control samples showed marked inflammation, which decreased in tumor-adjacent tissue by 30% but moderately increased in PC specimens by 54%. By comparing the type of immune cells (macrophage, lymphocyte, neutrophil ratio), RNASEL was significantly expressed only in control healthy specimens associated with lymphocyte infiltration.

The study performed has some limitations, the main being a limited number of samples. However, even on the presented number of samples, the consistent conclusions, with regard to sensitivity, specificity, positive predictive value (PPV), negative predictive value (NPV), false-negative rate, and cutoff values, for Poly(A) deadenylase will be drawn. The analysis of enzyme activity in plasma and urine and in liquid biopsies will answer whether there would be any interest for possible noninvasive diagnostic utility and screening.

## 5. Conclusions

In conclusion, significantly upregulated Poly(A) deadenylase activity in PC tissue and tumor-adjacent tissue associated with microfocal carcinoma highlighted it as a promising RNA–protein bypass biomarker of prostate cancer development. RNASEL may predict lymphocyte infiltration and chronic inflammation of the prostate.

## Figures and Tables

**Figure 1 cancers-14-02239-f001:**
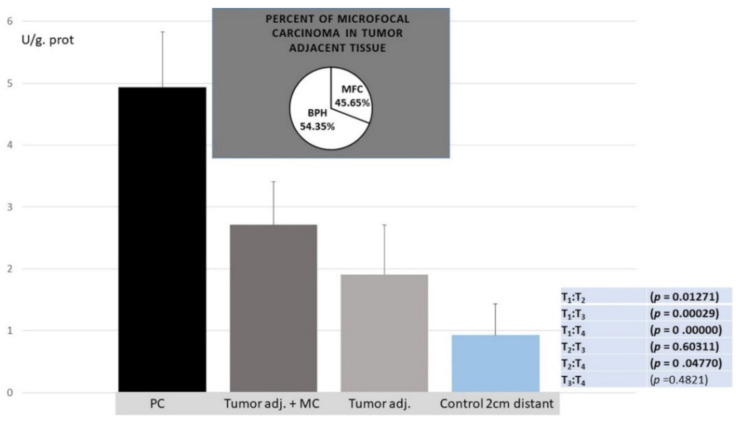
Poly(A) deadenylase specific enzyme activity (U/L g.prot) in PC, tumor adjacent with MC, tumor-adjacent and control healthy counterparts.

**Figure 2 cancers-14-02239-f002:**
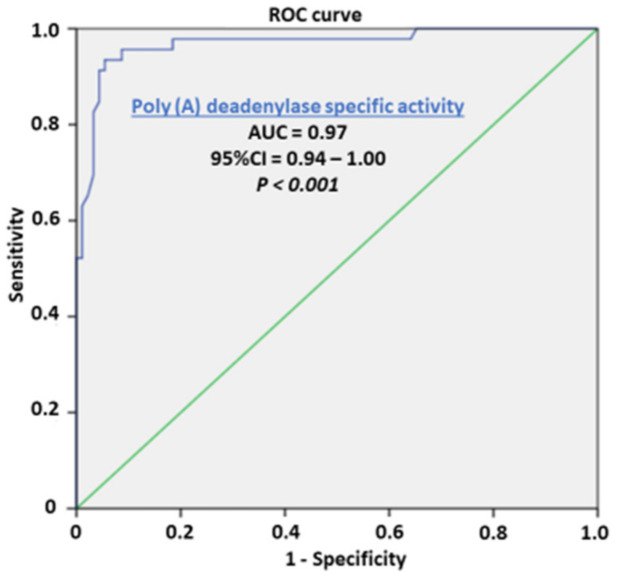
Receiver operating characteristic curve showing the performance of Poly(A) deadenylase specific activity in predicting prostate cancer.

**Figure 3 cancers-14-02239-f003:**
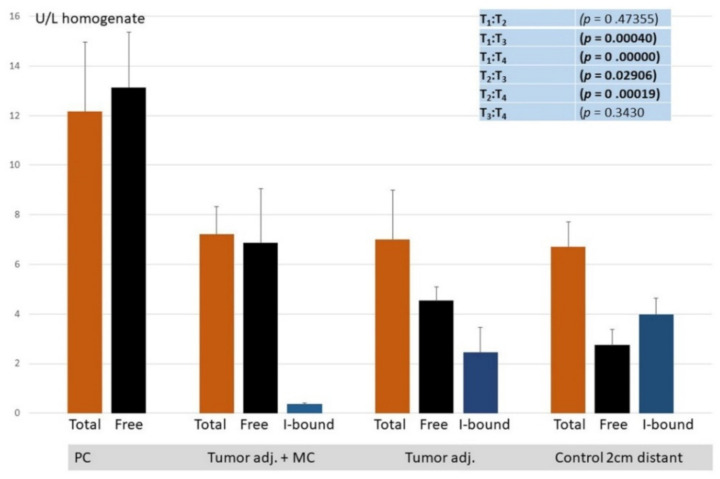
Total, free, and inhibitor-bound (latent) Poly(A) deadenylaseactivity(U/L) in PC, tumor adjacent with MC, tumor-adjacent, and control healthy counterparts.

**Figure 4 cancers-14-02239-f004:**
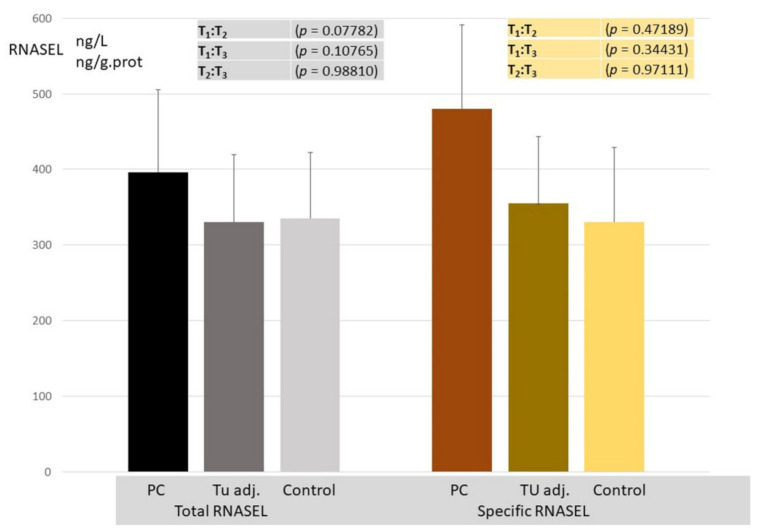
RNASEL total (ng/L) and specific (ng/g.prot) expression level in PC, tumor-adjacent, and control healthy counterparts.

**Figure 5 cancers-14-02239-f005:**
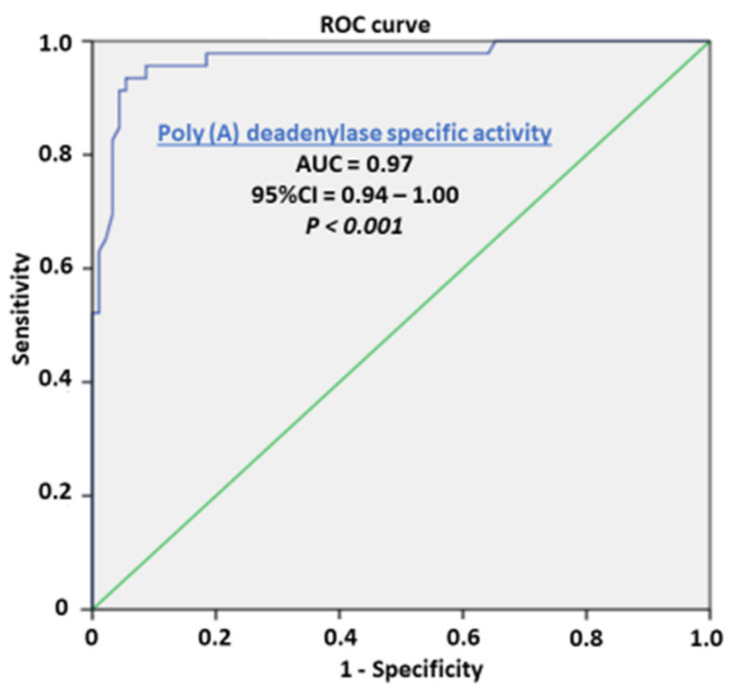
Receiver operating characteristic curve showing the performance of RNASEL-specific expression in predicting prostate cancer.

**Figure 6 cancers-14-02239-f006:**
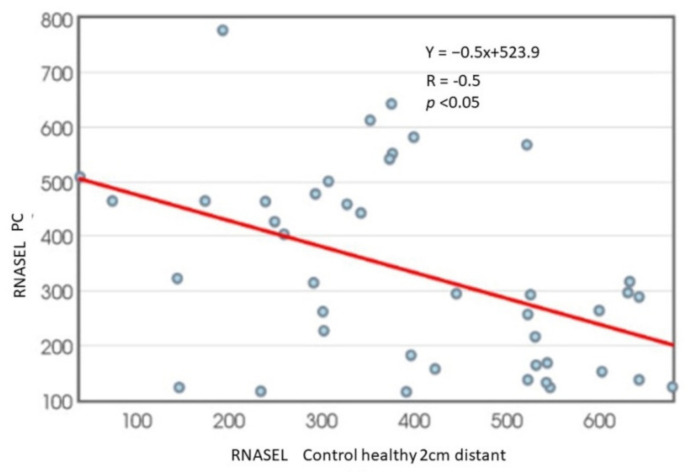
RNASEL correlation between values in PC and corresponding control specimens of healthy tissue.

**Figure 7 cancers-14-02239-f007:**
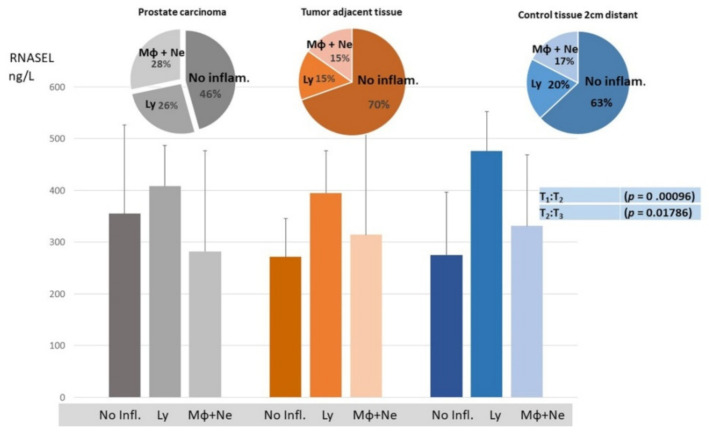
RNASEL in PC, tumor-adjacent and corresponding healthy tissues in relation to the presence and the type of inflammation.

**Figure 8 cancers-14-02239-f008:**
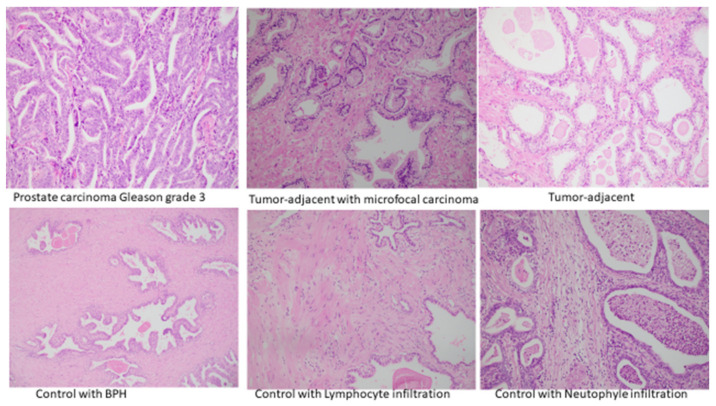
Histological findings of PC, tumor-adjacent tissue with microfocal carcinoma, tumor-adjacent, and control healthy tissue specimens.

**Table 1 cancers-14-02239-t001:** The age and level of PSA, Gleason score, and tumor stage in investigated patients with PC.

Investigated Parameters	n%
Age	
<70	28 (60.87%)
>70	18 (39.13%)
Tumor stage	
II	34 (73.91%)
III	12 (26.09%)
pN—lymph node metastasis	
NO	20 (43.48%)
NX	26 (56.52%)
pM-distant metastasis	
MO	22 (47.83%)
MX	24 (52.17%)
Gleason score	
3 + 3	15 (32.6%)
3 + 4	18 (39.13%)
4 + 3	9 (19.56%)
3 + 5	2 (4.35%)
4 + 4	2 (4.35%)
PSA (ng/mL)	
<10	28 (60.87%)
>10	18 (39.13%)

## Data Availability

All data are available in the personal files of BIOBANK patients and results, completed according to the project “IDEAS” needs.

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
