# Peer review of "Template-Independent Poly(A)-Tail Decay and RNASEL as Potential Cellular Biomarkers for Prostate Cancer Development"

_cancers, 2022, doi:10.3390/cancers14092239_

Round 1
Reviewer 1 Report
Data in table 1. do not make any sense regarding Gleason score and tumor stage.
Gleason score is usually expressed as 3+3, 3+4, 4+3, 4+4, 4+5 etc.
Tumor stage 2,27 , what does it mean?
Table 1. The age and the level of PSA, Gleason score and Tumor stage in investigated patients...
PC. 161
Investigated parameters X+SD
Age 68.27+ 3.45
(range 25–74 years)
tPSA (ng/mL) 15.84+8.77
Gleason score 6.76+0.59
Tumor stage 2.27+0.70
Author Response
Please find attached the reviewers answer and correction

Reviewer 2 Report
RNase L is a uniquely regulated endoribonuclease that requires 5'-triphosphorylated, 2',5'-linked oligoadenylates for its activity. RNase L functions in counteracting prostate cancer by virtue of its ability to degrade RNA, thus initiating a cellular stress response that leads to apoptosis. Intervening the role RNaseL in prostate cancer has been documented earlier. The author identified an increase of poly(A) deadenylase specific activity ranging from two to 247 ten times, followed by more than the twofold increase of its activity in adjacent carcinoma 248 tissue, compared to the control healthy tissue counterparts which are significant in this case. It will be worth calculating the sensitivity and specificity with respect to the above patient cohorts.
Author Response
Please find attached our answer and corrections.

Round 2
Reviewer 1 Report
Nicely corrected.
Reviewer 2 Report
The author addressed the raised concern